# A Survey of Optimizing ICU Sepsis Treatment Techniques in Reinforcement Learning Conference Submissions

## Abstract

Sepsis is a life-threatening condition that affects millions of people worldwide each year, characterized by high mortality and complex clinical trajectories. To address these complexities and the demand for real-time decision-making in clinical practice, RL(Reinforcement Learning), which emphasizes sequential decision-making and long-term reward maximization, has emerged as a promising approach, and numerous studies have sought to apply RL to optimize sepsis treatment. However, to date, no comprehensive survey has systematically analyzed the achievements and limitations of RL in sepsis care.

To bridge this gap, this paper (1)reviews the research landscape on RL-based approaches to sepsis treatment, (2)examines unresolved challenges and fundamental limitations of RL methods, (3)surveys recent technical advances designed to overcome these limitations and evaluates their strengths and weaknesses, and (4)proposes strategies for translating these methods into real-world clinical applications for sepsis management.

In conclusion, this study synthesizes the current state and limitations of RL-based sepsis treatment research, underscores the necessity of multi-layered approaches to address these challenges, and highlights future directions, particularly the introduction of Agentic AI systems capable of moving beyond simple treatment recommendations toward autonomous planning and execution.

## 1 Introduction

Sepsis is a life-threatening condition that threatens millions of lives worldwide each year, characterized by high mortality rates and complex clinical trajectories. Triggered by a systemic hyperinflammatory response to infection, sepsis can rapidly progress to septic shock and multiple organ failure if timely treatment is missed, often leading to fatal outcomes. Due to this clinical complexity, determining the optimal treatment strategy in real time for each individual patient remains one of the most challenging problems in critical care. Over the past decades, the international guideline for sepsis management, SSC(the Surviving Sepsis Campaign)(Evans et al., 2021), has been continuously revised. However, clinical trial results vary across patient cohorts, leading to inconsistent effectiveness of standardized treatment strategies, and large-scale randomized controlled trials face ethical and practical constraints.

Against this backdrop, Reinforcement Learning (RL) has attracted attention as an approach to support clinical decision-making in sepsis treatment by automatically learning optimal strategies from retrospective patient data. RL resembles the way physicians adjust treatments in response to changes in patient conditions, optimizing sequential decision-making processes to maximize outcomes. A study published in Nature Medicine in 2018(Komorowski et al., 2018) demonstrated the initial clinical applicability of RL through the 'AI Clinician' model trained on ICU sepsis patient data. Since then, numerous studies have extended this

approach, formalizing sepsis treatment as an RL problem and exploring diverse RL techniques to discover optimal strategies. However, to date, no comprehensive survey has systematically assessed the achievements and limitations of RL in sepsis treatment.

This paper aims to fill this gap by systematically analyzing prior research on RL-based sepsis treatment, offering the following key contributions. First, it provides a comprehensive review of RL applications in sepsis care in terms of data, MDP design, core algorithms, and evaluation methodologies. Second, it identifies fundamental limitations commonly faced across studies, including challenges in data, algorithms, evaluation, and safety. Third, it examines recent technical advances proposed to overcome these limitations and discusses how they enhance practical utility. Finally, it proposes the introduction of Agentic AI systems that move beyond mere prescription recommendations to autonomously planning treatment trajectories that incorporate long-term patient outcomes, thereby outlining the future direction of AI in sepsis care. **Summary of Contributions.** (1) This work presents the first systematic survey that reviews and analyzes research applying RL to sepsis treatment, and (2) it is the first paper to formally propose the Agentic AI paradigm for sepsis care.

## 2 BACKGROUND

### 2.1 CLINICAL BACKGROUND

**Definition and identification of sepsis**: Sepsis is defined as life-threatening organ dysfunction caused by a dysregulated host response to infection (Seymour et al., 2016). The earlier 1991 SIRS-based definition (Muckart & Bhagwanjee, 1997) showed poor specificity, so Sepsis-3 (2016) adopted the SOFA score (Vincent et al., 1996) to center the definition on organ failure. Sepsis is diagnosed when the SOFA score increases by 2 or more points, associated with hospital mortality above 10%. Septic shock is identified when VP(vasopressors) are required to maintain MAP(Mean Arterial Pressure) $\geq$ 65 mmHg despite adequate volume, together with serum lactate $\geq$ 2 mmol/L. This state is linked to in-hospital mortality above 40%. For bedside screening, qSOFA is positive if at least two of the following are present: respiratory rate $\geq$ 22/min, altered mentation, or systolic blood pressure $\leq$ 100 mmHg. However, the 2021 SSC guideline advises against using qSOFA alone for diagnosis.

**Treatment strategies in the ICU**: Sepsis and septic shock require urgent treatment. Key interventions include early fluid resuscitation (about 30 mL/kg crystalloids within 3 hours), VP(norepinephrine) if hypotension persists to maintain MAP $\geq$ 65 mmHg, and early broad-spectrum antibiotics, ideally within 1 hour in shock. Additional measures include source control (surgery or drainage), mechanical ventilation for respiratory failure, and renal replacement therapy for kidney failure.

**Patient status and clinical metrics**: In the ICU, severity is assessed through vital signs (blood pressure, heart rate, respiratory rate, temperature, oxygen saturation) and laboratory results (lactate, urine output, liver enzymes, inflammatory markers). Composite indices such as SOFA or APACHE II (APACHE, 1985) are often used. Primary outcomes include 28-day or in-hospital mortality, while secondary outcomes such as ICU/hospital length of stay and days alive without organ support better capture recovery pace even when mortality differences are small (Russell et al., 2018).

### 2.2 RL AND THE CONCEPT OF MDP IN A MEDICAL CONTEXT

Problems like sepsis management where a patient's condition evolves over time and clinicians must make sequential interventions can be modeled within the RL **MDP(Markov Decision Process)** framework(Komorowski, 2020). An MDP consists of five elements: the state space $S$, action space $A$, transition dynamics $P$, reward function $R$, and discount factor $\gamma$. Here, the state $s$ represents the patient's clinical condition; vital signs, lab results, and other physiological variables together constitute the state. An action

$a$ denotes the treatment decision available at that time for example, adjusting fluid infusion volume, titrating VP, modulating oxygen supplementation, or selecting antibiotics. When the agent observes the patient's state $s_t \in S$ at time $t$ and selects an action $a_t \in A$, the environment (the patient) transitions to the next state $s_{t+1}$ according to the probability $P(s_{t+1} \mid s_t, a_t)$, and a reward $r_t = R(s_t, a_t)$ is issued. The reward $r$ quantifies the immediate consequence of the action for the patient. The RL algorithm seeks a sequence of interventions (a policy) that maximizes long-term reward by choosing the most appropriate actions. In sepsis treatment, the ultimate objective is patient survival to hospital discharge, so it is common to assign a large positive terminal reward for survival and a large negative terminal reward for death. Because terminal outcomes are determined over the entire hospitalization, intermediate rewards are often designed to encourage improvements in the patient's condition during the episode.

The ultimate goal of RL is to learn a **policy** that maximizes cumulative reward through such interactions (policy optimization). Given a policy $\pi(a \mid s)$, we aim to maximize the expected discounted return $G_t = \sum_{k=0}^{\infty} \gamma^k r_{t+k}$, equivalently the action-value function $Q(s, a)$. The discount factor $\gamma \in [0, 1)$ balances the importance of immediate versus future rewards, encouraging the agent to consider both short-term responses and long-term survival. Policy optimization is broadly categorized into two families: **value-based** methods, which approximate $Q$ and select actions via $a = \arg\max_a Q(s, a)$, and **policy-based** and **actor–critic** methods, which directly optimize $\pi_\theta(a \mid s)$. Through these approaches, the agent continually improves its decision-making strategy, enhancing the system's ability to monitor and respond effectively to emerging health risks.

## 3 RESEARCH TRENDS AND LIMITATIONS OF RL-BASED SEPSIS TREATMENT

The comparison of studies applying RL to sepsis treatment can be found in Table 1 in the Appendix.

### 3.1 DATA FOUNDATIONS

#### 3.1.1 DATASETS AND CROSS-VALIDATION

**Major datasets:** RL-based sepsis treatment studies primarily utilize large-scale ICU clinical data. The most commonly used datasets are as follows. **MIMIC-III** (Johnson et al., 2016): It is a representative dataset developed by researchers from MIT and Beth Israel Deaconess Medical Center in Boston, containing detailed ICU patient records. **eICU** (Pollard et al., 2018): A large-scale multicenter ICU database provided by Philips Healthcare in collaboration with MIT, containing data from more than 200 hospitals across the U.S., enabling research in diverse clinical settings. Other publicly available datasets include **Amsterda-mUMCdb** (Thoral et al., 2021). Although these datasets are open access, users must complete registration, ethics/privacy training, and data use applications via platforms such as PhysioNet. However, it should be noted that while the most critical initial diagnosis and treatment decisions for sepsis patients occur in the ED(emergency department), the majority of existing studies have focused solely on ICU data. This limitation stems from the incompleteness and diagnostic uncertainty of ED data, resulting in RL models that cover only a small segment of the entire sepsis patient journey and thereby undermining their overall clinical validity(Nauka et al., 2025).

**Dataset Cross-Validation:** The aforementioned datasets are typically preprocessed and split into training, validation, and test sets for RL algorithm development and metric calculation. Most existing studies have relied on single-institution datasets for training and validation, raising concerns about whether models would perform equally well in previously unseen patient populations. Consequently, generalizability across hospitals and clinical settings remains limited. To address this, dataset cross-validation has been considered essential for verifying the robustness of RL models. However, differences in data recording practices and measurement units across hospitals make dataset integration difficult, hindering the effective use of external data and constraining model scalability and generalization. Some studies have attempted to extract common

variables and conduct cross-dataset training–evaluation (e.g., training on MIMIC and testing on eICU) to experimentally assess policy generalization. Yet, these efforts revealed performance degradation due to distributional shifts. While approaches such as FRL(Federated RL) have been explored to mitigate these gaps, such findings underscore that a single-policy or single-agent structure alone is insufficient to adequately absorb inter-institutional heterogeneity and temporal distributional changes(Oh et al., 2025). Therefore, a multi-agent architecture in which institution or expert-specific sub-agents learn local policies, and a higher-level coordinating agent integrates and mediates them is required to alleviate distributional shifts and enhance model generalizability

### 3.1.2 PREPROCESSING PROCEDURES AND LIMITATIONS

ICU raw data must undergo preprocessing before being used in RL model training, often with publicly available codebases (Microsoft Research, 2025), (MIT-LCP-mimic, 2018), (MIT-LCP-eICU, 2018). The main preprocessing steps are as follows:

**Patient selection and Episode construction:** Studies commonly select sepsis cohorts using the Sepsis-3 definition, identifying patients with a SOFA score increase of 2 or more and patient data are typically segmented around the diagnosis time and converted into state–action–reward trajectories, often with 4-hour intervals. To address variable sequence lengths, some studies truncate or pad episodes (Liang et al., 2023), (Do et al., 2020) whereas others allow variable-length episodes to better reflect patient trajectories (Choi et al., 2024). Fixed 4-hour windows may miss finer temporal dynamics; consequently, 1-hour (Lu et al., 2021) (Lu et al., 2020) and 2-hour (Wang et al., 2022) intervals have been explored, but the optimal temporal granularity remains an open question.

**Missing data handling:** Clinical datasets frequently exhibit $> 50\%$ missingness for certain variables. Common strategies include dropping variables with $> 70\%$ missingness, linear interpolation for low rates, and KNN-based imputation for intermediate rates. Komorowski et al. (2018) employ multivariable nearest-neighbor imputation, while Oh et al. (2025) use median substitution. However, many approaches ignore MNAR (Missing Not At Random) mechanisms, where missingness itself may signal clinical deterioration, thereby introducing bias(Rubin, 1976), (Little & Rubin, 2019).

**Data imbalance:** High mortality among severe cases yields imbalance between survival and death episodes, which can bias learning. Approaches such as undersampling or reweighting have been used(Tu et al., 2025); for example, randomly subsampling death episodes to balance survival and mortality cases can stabilize training.

**Normalization and outlier removal:** To mitigate scale bias, variables are log-transformed for long-tailed distributions or standardized with z-scores. Oh et al. (2025) normalize features to the $[0, 1]$ range, but such schemes are sensitive to outliers. Thus, clinically implausible or device-induced erroneous values are removed to prevent spurious patterns.

**Feature engineering:** Zhang et al. (2024) incorporate domain knowledge by deriving features such as SOFA and SIRS scores, whereas Lin et al. (2023) use auto-encoders to learn high-dimensional latent representations. Deep feature extraction, however, raises interpretability concerns and complicates clinical validation.

### 3.2 MDP DESIGN

RL-based studies on sepsis treatment formulate the patient's trajectory as a MDP, where patient states, clinical actions, and rewards are defined to learn an optimal treatment policy. The success of RL applications depends critically on appropriate design of state, action, and reward spaces, as well as on the choice of algorithms. The way the MDP is formulated directly impacts both model performance and clinical validity. Below we summarize how different studies have defined the components of the MDP.

### 3.2.1 STATE

Patient clinical status is high-dimensional and dynamically evolving. Failure to capture this complexity may undermine the consistency and generalizability of learned policies. Most studies derive patient states from EHRs(Electronic Health Records), using three main approaches. (1) The simplest "original" approach(Komorowski et al., 2018), (B"ock et al., 2022), (Drudi et al., 2024) uses raw clinical features observed at the current time step, or clusters patients into discrete groups used as state categories. This approach is intuitive but fails to capture temporal dynamics. (2) A second approach concatenates several recent time steps into a single state, Raghu et al. (2018) combined four consecutive measurements to incorporate short-term history. This captures trends but requires an arbitrary window size and increases dimensionality. (3) A more advanced approach uses temporal encoders to relax the Markov assumption and better capture patient dynamics: LSTMsLin et al. (2023) and RNNs(Raghu et al., 2018) map sequences to latent states, while auto-encoders(Raghu et al., 2017b), (Peng et al., 2018), (Do et al., 2020), (Lu et al., 2020), (Lu et al., 2021), (Raghu, 2019) or Transformers compress past $N$ hours of observations(Ma et al., 2023). Temporal encoding strengthens signal extraction from noisy EHRs but reduces interpretability and increases computation, and comparative studies show it yields the largest performance gains.

### 3.2.2 ACTION

Most studies discretize IVF(Intravenous Fluids) and VP dosages into quintiles, yielding $5 \times 5 = 25$ discrete treatment options. This simplifies learning by transforming continuous dosing into categorical decisions and mitigates data imbalance, but raises concerns regarding the clinical validity of five arbitrary bins and limits fine-grained dosing. To address this, Huang et al. (2022) adopt continuous action spaces with algorithms such as DDPG(Deep Deterministic Policy Gradient) or Twin-DDPG, directly predicting real-valued dosages. Safeguards are applied by penalizing implausible actions rarely taken by clinicians or by adding imitation terms to keep policies near observed distributions. However, critical treatment options such as antibiotic choice, oxygen supply, or ventilator settings are generally excluded, limiting clinical coverage. As a related example, the OptAB model(Wendland et al., 2024) predicts optimal antibiotic type and dosage using SOFA scores and pathogen information, but it optimizes a single-step decision and is not formulated as an RL model. Furthermore, most reward functions assume idealized scenarios without adverse events, failing to account for complications such as acute kidney injury or arrhythmias. Future work must expand the action space to incorporate drug type, dosing schedules, and patient-specific adjustments to achieve clinically comprehensive decision support.

### 3.2.3 REWARD

The reward function encodes clinical goals and is central to RL success. Since the ultimate objective is patient survival and recovery, many studies use terminal outcomes such as 90-day survival. However, sparse terminal rewards may destabilize training. To address this, intermediate rewards are often added, e.g., assigning positive rewards for SOFA score reduction or lactate clearance, and penalties for deterioration. Tu et al. (2025) use changes in Apache II scoresas intermediate signals, while Lu et al. (2020), Peng et al. (2018) replace survival with predicted log-odds of mortality at each step. Such differences in reward design shape learned policies differently, for instance, mortality-risk–based rewards may favor short-term stabilization, while SOFA-based rewards emphasize long-term organ function. To reduce uncertainty in hand-crafted reward design, Yu et al. (2019) used IRL(Inverse RL) to infer implicit reward functions from clinician trajectories. Aligning the reward with expert knowledge is seen as crucial for clinical adoption. Yet no consensus exists on the optimal reward formulation, and current designs rarely incorporate treatment side effects, ICU resource constraints, or economic costs. As a result, RL policies may recommend clinically infeasible strategies that overlook real-world limitations.

### 3.3 RL Methods

Research on RL for sepsis treatment has evolved from early Q-learning approaches to more advanced methods, including DRL, DistRL(Distributional RL), Conservative RL, IRL, and FRL. These approaches have progressively enabled more complex decision-making policies.

#### 3.3.1 RL paradigm: Offline RL

In healthcare, online RL where an agent explores novel strategies by interacting with real patients is infeasible due to ethical and safety concerns. Consequently, offline RL, which learns from retrospective clinical records, has become the standard approach (Tu et al., 2025). However, because offline RL cannot explore unseen states or actions beyond the dataset, agents are constrained to the range of historical clinician practices, limiting their ability to discover innovative strategies. Moreover, offline RL cannot reduce uncertainty through new interactions (Jayaraman et al., 2024), which raises challenges when actions are underrepresented in the data. Rare or unobserved actions are often overestimated in Q-values, leading to distributional shift and extrapolation errors. This creates the risk of recommending unvalidated treatments. To mitigate such risks, conservative RL methods restrict the policy search space or penalize unobserved actions to enhance safety.

#### 3.3.2 Core RL methods: Model-free vs. Model-based

***Model-free methods***. ***(1)Value-based approaches***: DQN(Ebrahimi & Lim, 2021) and its variants such as DDQN (Liu et al., 2020), Dueling DQN(Roggeveen et al., 2021), and D3QN are used to approximate the state-action value function to identify optimal discrete actions. Ruichang et al. (2022), Wu et al. (2023) proposed WD3QNE, which introduces dynamic weighting to balance the overestimation of Dueling DQN and the underestimation of D3QN. ***(2)Policy-based approaches***: For example, Lin et al. (2023) applied DDPG to continuous action spaces for dose optimization, reporting higher training efficiency and closer alignment with clinician decisions compared to DQN-based models.

***Model-based methods***. It has also been explored (Komorowski et al., 2018), (Raghu et al., 2018), (Wang et al., 2022). These approaches train environment simulators to model patient state transitions, allowing policy search within a virtual environment. Wang et al. (2022) combined behavioral cloning to initialize the policy with clinician actions, followed by PPO(Proximal Policy Optimization) refinement in the simulator. While model-based methods can improve data efficiency, errors in the learned environment model can negatively impact policy reliability.

#### 3.3.3 Hybrid methods

***DistRL:*** By modeling the full return distribution, DistRL captures uncertainty in outcomes (B"ock et al., 2022). Unlike standard Q-learning, which models only expected returns, DistRL estimates survival probability distributions, improving both performance and interpretability. However, calibration instability remains a challenge.

***Conservative RL:*** Algorithms such as CQL(Conservative Q-Learning) explicitly downweight the value of rarely observed or extreme actions to avoid overestimation, leading to safer policies that resemble clinician distributions (Tu et al., 2025),(Nambiar et al., 2023),(Kaushik et al., 2022),(Yu & Huang, 2022). Yet, excessive conservatism may over-constrain policies to past practices.

***IRL:*** Some studies used IRL to infer latent reward functions from clinician trajectories (Yu et al., 2019), (Yu & Huang, 2023). Especially, Yu & Huang (2023) analyzed treatment policy differences across race and gender, showing IRL's utility for understanding medical decision-making and fairness assessment beyond treatment optimization.

**FRL:** To enable multi-institutional collaboration and preserve data privacy, FRL has been proposed (Oh et al., 2025). Hospitals train local models and share only model parameters. FRL policies achieved performance comparable to centralized training, though heterogeneity across institutions poses generalization challenges.

**MoE(Mixture-of-Experts):** Hybrid architectures have combined kernel-based RL with DRL (Peng et al., 2018), or switched between supervised models(MLP) and RL models(DDQN) (Do et al., 2020). While effective, these approaches reduce interpretability due to opaque expert-switching mechanisms.

### 3.4 EVALUATION METHODS

Evaluation of RL-based sepsis treatment systems relies on **cross-dataset validation** for reliability and on both **quantitative** and **qualitative** assessments to examine policy performance and clinical validity from multiple angles.

#### 3.4.1 QUANTITATIVE EVALUATION

**OPE(Offline Policy Evaluation).** Because learning is offline, OPE is used to estimate policy value before any prospective deployment. Common estimators of expected return include IS(Importance Sampling), WIS(Weighted IS), and DR(Doubly Robust). IS/WIS estimate policy value by reweighting trajectories according to the probability that the new policy would have selected the logged actions, whereas DR corrects model bias by combining outcome modeling with IS. In Raghu (2019), variants such as per-horizon WIS/WDR have been applied to estimate policy $Q$-values and compare them against clinician policies. Tu et al. (2025) used FQE(Fitted $Q$ Evaluation), which trains a separate $Q$-function to evaluate a fixed policy. Using multiple OPE estimators improves confidence without patient risk, but uncertainty remains high for rare state–action pairs, and a single expected-value number can obscure patient-level heterogeneity.

**Clinical Outcome Metrics (Survival Rate).** Studies often report estimated survival under the learned policy, for example by comparing realized survival among cases where clinician care coincided with the RL recommendation versus where it did not, or by reporting absolute survival gains (percentage points) under the counterfactual policy. These are quasi-experimental estimates and are best interpreted as directional evidence of policy improvement.

#### 3.4.2 QUALITATIVE EVALUATION

To complement numeric metrics, qualitative analyses focus on medical plausibility and policy behavior. *Action-distribution* plots check whether the policy avoids extreme dosing or excessive conservatism and whether recommended actions resemble clinician prescribing patterns (*policy–clinician agreement*). High agreement can indicate safety, though it does not guarantee superior outcomes. Finally, *case studies* trace state–action–outcome trajectories, contrasting scenarios where clinicians followed versus deviated from the RL recommendation; such analyses illustrate whether the policy aligns with clinical reasoning and whether adherence corresponds to improved outcomes.

## 4 INTRODUCTION OF AGENTIC AI SYSTEMS: TOWARDS A NEXT-GENERATION TREATMENT SYSTEM

To overcome the limitations of the RL approaches reviewed earlier, a new paradigm is required(Nauka et al., 2025). As an alternative, this paper proposes the introduction of an **Agentic AI system**. Agentic AI refers to autonomous AI systems designed to independently analyze information from the environment, make their own decisions, and perform complex tasks to enhance operational efficiency across domains(Rossi et al.,

2025). Unlike conventional RL, which recommends actions step by step based on fixed policies, Agentic AI is characterized by its ability to autonomously set long-term treatment goals, establish and execute multi-step plans, and revise those plans as necessary. Furthermore, Agentic AI goes beyond simply combining RL with LLMs(Large Language Models) by leveraging memory, planning, and tool-use capabilities to proactively adapt to complex clinical environments. Recent studies emphasize the potential of such domain-specific agents, showing that agents designed to interact with human experts, respond to real-time scenarios, and acquire relevant knowledge can reduce contextual learning errors and improve the accuracy and robustness of clinical decisions compared to general-purpose LLMs(Ruiz Mejia & Rawat, 2025). Below, we propose how Agentic AI can address the limitations of existing sepsis treatment research.

### 4.1 MULTI-STEP AUTONOMOUS PLANNING

Agentic AI can autonomously establish long-term treatment plans. For instance, the agent may construct a multi-step plan such as "stabilize vital signs within the first 6 hours, then monitor the patient and adjust drug regimens as needed." This goes beyond recommending only the optimal action at a single time point (as in RL) by planning a sequence of actions over time, thereby mitigating the challenges of sparse and delayed rewards. Moreover, the agent can flexibly adapt by revising plans in real time when patient conditions deviate from expectations. Such autonomy and self-correction are particularly valuable for rapidly evolving conditions like sepsis.

### 4.2 KNOWLEDGE INTEGRATION AND TOOL USE

Agentic AI can expand its action space and information access by connecting with external knowledge bases and tools. For example, the MATEC framework(Cho et al., 2025) utilizes RAG(Retriever-augmented Generation) to access resources such as the IDSA sepsis guidelines, Penn sepsis treatment guidelines, and Penn antibiotic guidelines, enabling the agent to design treatment plans grounded in the latest clinical knowledge. The agent can search for optimal antibiotic combinations or query specialized databases to assess risks of adverse drug reactions. Such tool use enables a broader treatment space, beyond fluid resuscitation or VP dosing, thus mitigating the limitations of narrow action spaces and unmeasured confounders in prior RL-based studies.

Moreover, Agentic AI can address limitations in missing data handling. Instead of mechanically applying threshold-based imputation, it can interpret missing patterns as clinical signals and detect the context of missingness. For instance, when certain lab tests are more often missing in high-risk patients, the agent may encode missingness as an intentional clinical signal (indicator feature), integrating it into risk prediction and treatment recommendation rather than simply imputing values.

Furthermore, this enables multimodal data integration. The agent can jointly interpret lab results, medical imaging, and genomic data, thereby improving state estimation accuracy by accounting for the multifaceted pathophysiology of sepsis. Ultimately, Agentic AI can implement knowledge-driven actions that alleviate the challenges of RL reward function design and dataset bias.

### 4.3 MULTI-AGENT COLLABORATION AND SCALABILITY

Given the complexity of sepsis care, multi-agent frameworks can be more effective, with specialized agents managing different aspects of patient care. For example, recent work(Shaik et al., 2023) introduced separate RL agents for monitoring key vital signs (heart rate, respiration, body temperature), thereby alleviating reward sparsity and improving learning efficiency. In such systems, agents typically collaborate by sharing patient information and reward signals, leading to improved system-wide performance. For instance, one agent may manage fluid therapy while another handles VP dosing, coordinating interventions for faster and more balanced treatment. In resource-limited settings, agents may switch to competitive modes, prioritiz-

ing the most critical patients and adapting alarm urgency. Importantly, modular design enables scalable extension new agents (e.g., for additional physiological variables) can be integrated without performance degradation. This flexibility supports deployment in large-scale hospital monitoring scenarios. Multi-agent structures can also enhance missing data handling: parallel agents may employ diverse strategies, while a coordinating agent selects the optimal imputation based on uncertainty, clinical plausibility, and consistency. This minimizes information loss and bias arising from missing data.

### 4.4 EXPLAINABILITY AND TRUST

Unlike black-box RL policies, Agentic AI systems offer enhanced interpretability through reasoning traces and communication capabilities. For example, the agent may record its decision rationale in a chain-of-thought format or visualize attention weights to highlight patient features influencing treatment recommendations. Such transparency is critical in clinical domains where decision-making must be auditable(Brohi et al., 2025). By providing explanations of recommendations and expected outcomes, Agentic AI can foster clinician trust. Safety is further enhanced by embedding predefined medical rules (e.g., dosage limits, contraindications), ensuring the agent avoids harmful actions. When violations occur, the system can issue alerts or propose alternatives, allowing human experts to correct errors collaboratively.

### 4.5 HUMAN-AI COLLABORATION AND PRACTICALITY

The proposed Agentic AI system is designed to complement, not replace, clinicians. Evolving beyond conventional decision-support systems, it acts as an intelligent collaborative partner. For example, it can analyze complex data streams in real time, propose optimal treatment pathways, and provide supporting rationales, while leaving final decisions to physicians. In sepsis where multidisciplinary expertise is essential the agent can rapidly reference relevant guidelines or retrieve lessons from past cases, enriching clinical judgment. This synergy combines AI's computational strengths with physicians' intuition and experience. Additionally, by continuously monitoring patients and detecting subtle changes, Agentic AI can alleviate clinician workload and ensure continuity of care. Ultimately, such systems aim not only to recommend treatments but also to autonomously plan, execute, and interact with clinicians as next-generation care frameworks. Real-world deployment will require rigorous validation, phased clinical trials, and cross-disciplinary collaboration among clinicians, data scientists, and safety engineers.

## 5 CONCLUSION

Sepsis remains a major global health challenge due to its high mortality rate and complex clinical course, and research efforts applying RL to address this problem have rapidly evolved. This paper provides a comprehensive review of existing RL-based studies in sepsis treatment, offering a multilayered analysis of their achievements and limitations. While prior work has demonstrated that RL can learn optimal treatment policies from clinical data and potentially suggest strategies superior to those of clinicians, key barriers to clinical adoption remain. These include dataset limitations, incomplete MDP design, sparse and uncertain rewards, restricted action spaces, poor generalizability, as well as insufficient interpretability and safety. We highlight that these issues are structural and fundamental constraints, unlikely to be resolved by algorithmic improvements alone.

To overcome these challenges, we propose the adoption of Agentic AI as a next-generation paradigm. Agentic AI integrates RL's optimization capability with the reasoning and tool-use abilities of LLMs, moving beyond one-step treatment recommendations toward setting long-term therapeutic goals, formulating multi-stage plans that can be revised in response to changing patient conditions, and leveraging external knowledge resources. Such autonomy and adaptability address limitations of RL including data bias, reduced action

spaces, and lack of interpretability while enhancing safety and trustworthiness through collaboration with clinicians.

The introduction of Agentic AI, capable of actively reasoning about patient states and continuously updating optimal treatment strategies, is anticipated to transform the paradigm of sepsis care, substantially improving survival outcomes and optimizing the use of medical resources. Ultimately, Agentic AI has the potential to evolve into a reliable clinical decision-making partner in sepsis care. By dynamically monitoring patient status, providing treatment pathways aligned with long-term objectives, and alleviating the workload of healthcare providers, it could become a cornerstone technology in critical care. We therefore argue that future research on sepsis treatment should focus on integrating the foundational progress of RL with the Agentic AI framework in a multilayered approach, thereby paving the way for safe and practical adoption in clinical practice.

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

# A  APPENDIX

Table 1: Comparison of papers using RL in the treatment of sepsis

| PAPER | MODEL | DATASET | STATE | ACTION | REWARD | METRICS | NOTES |
|---|---|---|---|---|---|---|---|
| (Oh et al., 2025) | FRL (FedAvg, FedProx) + Highlight D3QN | MIMIC-III ≈ 257,162 records; eICU v2.0 ≈ 183,040 records (4h intervals) | 47 clinical variables; continuous state | IV & VP, 5 levels each ≈ 25 discrete actions | Final (90-day survival): 1 | Action Distribution Survival-dose gap Estimated Mortality/Survival | Federated learning approach for privacy preservation |
| (Tu et al., 2025) | CQL | MIMIC-III ≈ 14,957 sepsis patients; 380,456 time-series records; Avg. per-patient length 100h (SD 88h) | 48 clinical variables; continuous state space | IV & VP, 5 levels each ≈ 25 discrete actions | Intermediate: Apache II?ased reward; Final (90-day survival): 1 | Expected Return; Clinical Policy Similarity | `https://github.com/OOPSDINOSAUR/RL_safety_model` |
| (Liang & Jia) | RL-NN | MIMIC-III ≈ 412 sepsis patients | Stage 1: 7 demographics; Stage 2: 9 vars (Stage 1 + treatment outcome) | Three fluid dosing tiers: <20, 20–0, >30 mL/kg | SOFA-based: $Y = \exp\{25 \approx SOFA/17\}$ | Percent optimal F1 SOFA reduction Precision Recall Treatment allocation ratio | Two-stage policy (0≈ h; 3–4h); includes insurance & religion; simulation with 5/20/50 synthetic vars to assess %opt |
| (Zhang et al., 2024) | Safe-D3QN | MIMIC-III ≈ 19,582 sepsis patients | 46 clinical variables; continuous state | IV & VP, 5 levels each ≈ 25 discrete actions | Compared 3 rewards; adopted SOFA & lactate reduction | Expected Return; Estimated Mortality/Survival | Safety constraints/penalties for abrupt dosage changes; more gradual VP use |
| (Drudi et al., 2024) | CARDIO | MIMIC-III ≈ 20,496 sepsis patients; 72h (?4h to +48h); 4h intervals | FULL variant: 48 vars clustered (k-means++) ≈ 750 discrete states + survival label | IV & VP, 5 levels each ≈ 25 discrete actions | Final (90-day survival): 100 | Expected Return | Built RL recommender using only cardiopulmonary variables (e.g., HR, BP); compared FULL/CARDIO/PCA; FULL ≈ 750+2 states |

Table 1: Comparison of papers using RL in the treatment of sepsis

| PAPER | MODEL | DATASET | STATE | ACTION | REWARD | METRICS | NOTES |
|---|---|---|---|---|---|---|---|
| (Wang et al., 2024) | SAI-DQN | MIMIC-IV $\approx$ 9,982 sepsis patients; eICU $\approx$ 11,070 sepsis patients | 311 variables (12 antibiotics, 26 labs, 4 vitals, demographics, cultures); continuous state | 30 antibiotic regimen clusters (k-means) incl. dosing duration | R1: 90-day outcome 100; R2: SOFA change 10; R3: Treatment duration >9 days $\approx$ 0; R4: Clinical realism term (Qclinical0.1) | Expected Return Precision Recall Clinical Policy Similarity | Guideline knowledge integrated into reward |
| (Choi et al., 2024) | Highlight D3QN | Training: MIMIC-III $\approx$ 20,927 sepsis patients; 14,957 trajectories; Validation: eICU v2.0 $\approx$ 14,875 patients; first 80h, 4h intervals | 47 clinical variables; continuous state space | IV & VP, 5 levels each $\approx$ 25 discrete actions | Final (90-day survival): 1 | Estimated Mortality/Survival | `https://zenodo.org/records/13842300` |
| (Tamboli et al., 2024) | POSNEGDM + Transformer | MIMIC-III $\approx$ 19,614 trajectories; 4h intervals | Estimated Mortality/Survival; Clinical Policy Similarity | | Final (end-of-trajectory survival): 1 | $\approx$ 17 clinical variables; continuous state | |
| (Wu et al., 2023) | WD3QNE | Training: MIMIC-III $\approx$ 17,083 patients; 276,232 records. Validation: eICU $\approx$ 1,500 patients; 24,279 records; 80h; 4h intervals | 37 variables (selected via Random Forest from 45); continuous state | IV & VP, 5 levels each $\approx$ 25 discrete actions | Intermediate: SOFA-based; Final (90-day survival): 24 | Action Distribution; Expected Return; Estimated Mortality/Survival | `https://github.com/CaryLi666/ID3QNE-algorithm` |
| (Lin et al., 2023) | DDPG | MIMIC-III $\approx$ 38,600 time-series records; within 72h after onset; 4h intervals | Estimated Mortality/Survival; Clinical Policy Similarity; Training Efficiency | IV & VP, 5 levels each $\approx$ 25 discrete actions | Intermediate: SOFA & lactate reduction; Final survival: 15 | Expected Return | $\approx$ 48 variables $\approx$ autoencoder $\approx$ 200D continuous state |

Table 1: Comparison of papers using RL in the treatment of sepsis

| PAPER | MODEL | DATASET | STATE | ACTION | REWARD | METRICS | NOTES |
|---|---|---|---|---|---|---|---|
| (Ma et al., 2023) | DAQN | MIMIC-III 1.4 ≈ 6,164 sepsis patients; 4h intervals | Attention-based embeddings of clinical variables; continuous state | IV & VP, 5 levels each ≈ 25 discrete actions | SOFA & lactate reduction | Expected Return | Transformer-style attention encodes observations & action history |
| (Nambiar et al., 2023) | CQL | MIMIC-III ≈ 18,923 trajectories; 4h intervals | 44 clinical variables; continuous state | IV & VP, 5 levels each ≈ 25 discrete actions | ICU survival label every 4h up to 48h: 1 | Expected Return; Clinical Policy Similarity | Offline RL for treatment optimization |
| (Bologheanu et al., 2023) | Actor-critic RL | AmsterdamUMCdb ICU ≈ 2,946 sepsis patients; 3,051 trajectories; 24h window | 379 clinical variables (excluding mortality & steroid dose); continuous state | Five discrete corticos-teroid dose bins | Positive reward for ICU survival/discharge; negative for ICU death; stepwise rewards after actions | Action Distribution; Expected Return; Estimated Mortality/Survival; Clinical Policy Similarity | Optimizes corticosteroid dosing |
| (Yu & Huang, 2023) | D3QN | MIMIC-III v1.4 ≈ 14,012 sepsis patients | 30D continuous state space | IV & VP, 5 levels each ≈ 25 discrete actions | SOFA & lactate reduction; integrates PaO2 & PT via MiniTree (MT) to align short- & long-term outcomes | Expected Return; Estimated Mortality; Treatment Efficiency | Balances short-term improvement with long-term mortality signals |
| (Liang et al., 2023) | D3QN | MIMIC-III ≈ 17,898 sepsis patients; fixed-length; 4h intervals | 48 clinical variables; continuous state | IV & VP, 5 levels each ≈ 25 discrete actions | Intermediate: SOFA & lactate (C0=?.025, C1=?.125, C2=?); Final (discharge alive): 15 | Expected Return; Estimated Mortality/Survival | Stores similar past episodes as episodic memory; https://github.com/DMU-XMU/Episodic-Memory-assisted-Approach-for-Sepsis-Treatment |

Table 1: Comparison of papers using RL in the treatment of sepsis

| PAPER | MODEL | DATASET | STATE | ACTION | REWARD | METRICS | NOTES |
|---|---|---|---|---|---|---|---|
| (Ruichang et al., 2022) | WD3QNE | MIMIC-III v1.4 ≈ 17,083 sepsis patients; 276,232 records; 4h intervals | 37 variables (from 45 via RF); continuous state | IV & VP, 5 levels each ≈ 25 discrete actions | Intermediate: SOFA-based; Final (90-day survival): 24 | Action Distribution; Expected Return; Estimated Mortality/Survival | Addresses limits of discrete state spaces by using continuous variable selection |
| (Kaushik et al., 2022) | CQ network policy | MIMIC-III v1.4 ≈ 4h intervals | Action Distribution; Estimated Mortality/Survival | IV & VP, 5 levels each ≈ 25 discrete actions | Intermediate: SOFA & lactate reduction; Final (end-of-trajectory survival): 15 | IV & VP, 5 levels each ≈ 25 discrete actions | ≈ 48 clinical variables; continuous state |
| (Yu & Huang, 2022) | PAS curriculum offline RL | MIMIC-III v1.4 ≈ 22,095 sepsis patients; 72h (?4h to +48h); 4h intervals | 46 clinical variables; continuous state | IV & VP ≈ continuous action space | Final (end-of-trajectory survival): 15 | Expected Return; Estimated Mortality/Survival | Curriculum training with progressively expanded action space |
| (Wang et al., 2022) | LSTM; VAE; CDQ | MIMIC-IV ≈ 6,660 patients meeting Sepsis-3 within first 24h; 2h intervals | 41 clinical variables; continuous state | Continuous IV; 3 VP categories; discrete hydrocortisone use | Intermediate: SOFA & lactate reduction; Final (end-of-trajectory survival): 25 | Estimated Mortality/Survival; Clinical Policy Similarity | Model-based RL |
| (Huang et al., 2022) | DDPG, TD3 | MIMIC-III ≈ 19,633 sepsis patients; 4h intervals; 84h records | 38 normalized clinical variables; continuous state space | IV & VP ≈ continuous action space | SOFA-based reward | Action Distribution; Expected Return | Code reported invalid/unreproducible |
| (B"ock et al., 2022) | Categorical DQN; Speedy Q-learning | MIMIC-III ≈ 957,563 time-series records | Expected Return; Estimated Mortality/Survival | | Single terminal reward (28/90-day survival): 100 | Reduced discrete action space (pruned from 25) | ≈ 53 variables clustered with k-means ≈ 19 discrete states |
| (Jia et al., 2021) | DQN | MIMIC-III ≈ 25,247 sepsis patients | 47 clinical variables; continuous state | IV & VP, 5 levels each ≈ 25 discrete actions | Intermediate: SOFA & lactate reduction; Final (90-day survival): 15 | Dosage change rate (safety) | `https://github.com/Yanjiayork/sepsisRL` |

Table 1: Comparison of papers using RL in the treatment of sepsis

| PAPER | MODEL | DATASET | STATE | ACTION | REWARD | METRICS | NOTES |
|---|---|---|---|---|---|---|---|
| (Lu et al., 2021) | D3QN | MIMIC-III v1.4 ≈ 2,492 septic shock patients receiving VP & IV; mean 24h; 1h intervals; 59,503 records | 42 variables encoded by LSTM autoencoder | IV & VP, 5 levels each ≈ 25 discrete actions | Reward encourages MAP 65–5 mmHg; penalties for excess IV/VP | Expected Return; Estimated Mortality/Survival | Analyzes sensitivity to including cumulative IV/VP history; evaluates action distribution rather than mortality |
| (Lu et al., 2020) | D3QN | MIMIC-III v1.4 ≈ 7,956 patients; 649,661 records; 1h intervals | 52 variables encoded by LSTM autoencoder (fixed-length summary incl. cumulative history) | IV & VP, 5 levels each ≈ 25 discrete actions | Short-term reward: 30-day mortality negative log-odds change; Long-term reward: 1-year survival + discharge SOFA | Clinical Policy Similarity | Compares policies under short- vs long-term reward definitions; focuses on action distribution |
| (Do et al., 2020) | DDQN + supervised learning | MIMIC-III v1.4 ≈ 17,928 sepsis patients; 4h intervals; fixed spacing per patient | 42 variables ≈ latent continuous state | IV & VP, 5 levels each ≈ 25 discrete actions | Intermediate: SOFA & lactate reduction; Final (end-of-trajectory survival): 15 | Expected Return; Clinical Policy Similarity | Sparse autoencoder latent state |
| (Jia et al., 2020) | DQN | MIMIC-III ≈ 25,247 sepsis patients | 47 clinical variables; continuous state | IV & VP, 5 levels each ≈ 25 discrete actions | Intermediate: SOFA & lactate reduction; Final (90-day survival): 15 | Expected Return; Dosage change rate | Motivation for 90-day mortality over in-hospital due to discharge bias |
| (Roggeveen et al., 2021) | D3QN | Training: MIMIC-III v1.4 ≈ 72h (?4h to +48h); Validation: Amsterdam-UMCdb ≈ 72h from admission | 43 features; (state representation not further specified) | 21 discrete actions (subset of IV/VP 55 grid) | Final (end-of-trajectory survival): 15 | Action Distribution; Expected Return | Clinical feasibility constraint: exclude 4 actions where IV=0 & VP>0 ≈ 21 valid actions |

Table 1: Comparison of papers using RL in the treatment of sepsis

| PAPER | MODEL | DATASET | STATE | ACTION | REWARD | METRICS | NOTES |
|---|---|---|---|---|---|---|---|
| (Yu et al., 2019) | DIRL-MT | MIMIC-III | 7 clinical variables; continuous state | Continuous dosing of IV & VP | Reward learned from clinician behavior and state transitions (IRL) | Estimated Mortality/Survival | Inverse RL approach |
| (Peng et al., 2018) | MoE | MIMIC-III (v1.4) ≈ 15,415 sepsis patients; 4h intervals | Expected Return | IV & VP, 5 levels each ≈ 25 discrete actions | Reward = change in mortality log-odds per step (range ~[-3, 3]) | | ≈ 50 clinical variables ≈ LSTM autoencoder to 128D continuous state |
| (Raghu et al., 2018) | NN PPO | MIMIC-III v1.4 ≈ 17,898 patients; 72h (24h pre to 48h post); 4h intervals | Current 48 vars concatenated with previous 3 steps ≈ 198D continuous state | IV & VP, 5 levels each ≈ 25 discrete actions | Intermediate: SOFA & lactate reduction; Final (90-day survival): 15 | Expected Return | Model-based RL |
| (Komorowski et al., 2018) | Policy Iteration | Training: MIMIC-III (17,083 sepsis patients); Validation: eICU (79,073 patients); 72h, 4h intervals | 48 variables clustered with k-means ≈ 750 discrete states | IV & VP, 5 levels each ≈ 25 discrete actions | Final (90-day survival): 100 | Expected Return | Pioneer ICU RL study; `https://github.com/matthieukomorowski/AI_Clinician` |
| (Raghu et al., 2017b) | Autoencode Q-N | MIMIC-III v1.4 ≈ 17,898 sepsis patients; 72h (?4h to +48h); 4h intervals | 47 variables ≈ autoencoder ≈ continuous state | IV & VP, 5 levels each ≈ 25 discrete actions | Final (90-day survival): 15 | Expected Return; Estimated Mortality/Survival | Motivation: lack of consensus on IV/VP dosing standards |
| (Raghu et al., 2017a) | D3QN | MIMIC-III v1.4 ≈ 17,898 sepsis patients; 4h intervals; 72h window (24h pre- to 48h post-diagnosis) | 48 clinical variables; continuous state space | IV & VP, 5 levels each ≈ 25 discrete actions | Intermediate: SOFA decrease & lactate reduction; Final (90-day survival): 15 | Action Distribution; Estimated Mortality/Survival | `https://github.com/aniruddhraghu/sepsisrl` |

## B  USE OF LARGE LANGUAGE MODELS (LLMS)

In accordance with the ICLR 2026 policy on the disclosure of LLM usage, we provide a detailed description of how LLMs were utilized during the preparation of this work.

### B.1  WRITING ASSISTANCE

We used an LLM (OpenAI's ChatGPT, GPT-5) to support the writing process. Specifically, the model was employed to:

- Improve clarity and coherence of sentences by suggesting alternative phrasings.
- Assist in editing for grammar, style, and consistency.

All content generated by the LLM was carefully reviewed, verified, and revised by the authors to ensure accuracy, originality, and compliance with scientific standards. The final responsibility for the content rests solely with the authors.

### B.2  RESEARCH SUPPORT

LLMs were also employed as an auxiliary tool to facilitate the discovery of relevant literature. This included:

- Identifying related works in RL and sepsis treatment.
- Organizing references for inclusion in the manuscript.

All references were independently validated by the authors using original publications to prevent errors, omissions, or misrepresentations.

### B.3  LIMITATIONS OF LLM USE

It is important to note that while the LLM assisted in writing and literature organization, it did not contribute original research ideas, data analysis, experimental design, or results generation. Therefore, the LLM is not listed as an author or co-author, in accordance with the ICLR 2026 policy.

