# OpenReview forum: "A Survey of Optimizing ICU Sepsis Treatment Techniques in Reinforcement Learning"
_ICLR.cc/2026/Conference — Submitted to ICLR 2026_

### Official Review · Reviewer_R5AV · 2025-10-31

**Soundness:** 2
**Presentation:** 2
**Contribution:** 2
**Rating:** 4
**Confidence:** 4

**Summary:**

This paper provides a comprehensive survey of reinforcement learning (RL) methods applied to sepsis treatment in intensive care. It reviews major datasets (MIMIC, eICU, AmsterdamUMCdb), MDP formulations, and the evolution from tabular and deep RL methods to more recent variants such as offline RL, federated RL, and conservative RL. The authors also propose Agentic AI as a future paradigm that integrates RL with reasoning, planning, and tool-use capabilities from large language models (LLMs), highlighting potential directions such as multi-agent coordination, knowledge integration, and explainability.

**Strengths:**

	The paper effectively summarizes key technical developments in RL for sepsis treatment, providing an extensive comparison table that will be useful for newcomers to this domain.
	The intersection of RL, healthcare, and LLM-based agents is of current interest to both the ML and clinical AI communities.

**Weaknesses:**

	In Section 2.1 (“Treatment Strategies in the ICU”), additional references should be included to support the listed treatment measures and indicate which studies specifically investigated their effects on sepsis outcomes. Besides, In Section 3.1.1 (“Dataset Cross-Validation”), please cite concrete works that attempted to extract common variables and conduct cross-dataset training–evaluation, as currently this statement is unsupported. Sections 3.4.1 and 3.4.2 also need proper citations for the studies being summarized.
	The discussion of limitations appears scattered across subsections in Section 3; consolidating these limitations into a dedicated section could make the analysis more coherent and insightful.
	The transition from the RL survey to the Agentic AI discussion is abrupt. The paper does not show how the latter concretely addresses the limitations identified earlier.
	While the survey is thorough, it largely restates prior works and their limitations. The proposed paradigm in Section 4 does not introduce new techniques or theoretical insights that would meet ICLR’s originality criteria. A detailed reproduction or meta-analysis such as comparing how different reward designs influence policy behavior across existing RL algorithms for sepsis could substantially enhance the scientific value and make the paper more aligned with ICLR’s expectations.

**Questions:**

	How does the introduction of Agentic AI in Section 4 concretely address the limitations identified in Section 3? It would be helpful to clarify which specific challenges the proposed paradigm aims to solve, and through what mechanisms or modeling extensions.
	Can the authors provide more analytical or empirical insights, beyond summarization, that help advance understanding of RL for sepsis treatment? For example, would it be feasible to replicate or compare existing studies to analyze the impact of different reward designs or environment assumptions on learned policies?

---

### Official Review · Reviewer_7oJs · 2025-10-31

**Soundness:** 2
**Presentation:** 3
**Contribution:** 2
**Rating:** 2
**Confidence:** 3

**Summary:**

This paper surveys how RL has been used to optimize ICU sepsis treatment, from data pipelines and MDP design to core algorithms and evaluation. It also argues that current offline RL approaches have structural limits in clinical practice (data bias, narrow action spaces, weak generalization, interpretability/safety gaps) and proposes a next-step paradigm—Agentic AI—that plans multi-step care, integrates guidelines/tools, collaborates with clinicians, and may better address those limits in real deployments.

**Strengths:**

1. Importance. The paper takes on sepsis decision-making with RL, it seems timely and clinically consequential.

2. Clarity. I like the presentation of the paper, it seems easy to navigate and well organized.

**Weaknesses:**

You state this is the first systematic survey of RL for sepsis and the first to formally propose Agentic AI for sepsis care; can you bound that claim to substantiate uniqueness?

As far as I know, the work Reinforcement Learning in Dynamic Treatment Regimes Needs Critical Reexamination by Luo et al, systematically re-examines RL for dynamic treatment regimes (incl. sepsis) and, via a large sepsis case study, highlights evaluation pitfalls, MDP/design inconsistencies, and baseline issues that overlap with—and some go beyond—your survey’s coverage.

Note that the authors did not cite this highly relevant work.

Furthermore, if you consider the so-called Agentic-AI paradigm a primary contribution, and argue it addresses structural limits (data bias, restricted actions, interpretability); this is unsound claim, as there is neither theoretical nor empirical support.

For the Agentic-AI proposal is, what’s the minimal novel technical commitment (e.g., planning module + RAG + rule-checks) that distinguishes it from “RL + guideline prompts,” and can you show a tiny worked example (antibiotics + fluids for 24h) to make that mechanism concrete?

The paper discusses expanding action spaces (e.g., antibiotics) and continuous dosing; can you highlight which studies already do this and propose a standard expanded-action benchmark checklist (variables, bins/continuous ranges, safety guards) to guide future work?

**Questions:**

see above.

---

### Official Review · Reviewer_DkcG · 2025-11-02

**Soundness:** 1
**Presentation:** 2
**Contribution:** 1
**Rating:** 0
**Confidence:** 5

**Summary:**

This paper provides a broad overview of techniques and approaches to using reinforcement learning for the clinical treatment of Sepsis. From the literature, the authors then identify key limitations that have arisen blocking actual deployments of proposed systems. Following this, the authors introduce a conceptual framework of agentic AI to address some of these limitations.

**Strengths:**

The paper does a good job creating a clear categorization of relevant concepts, algorithmic procedures, and important considerations taken to address sepsis treatment in RL research. This does a nice job outlining important factors that prior work has considered as well as lays bare limitations of these prior papers.

**Weaknesses:**

This paper, while earnest and well meaning, did not produce anything that hasn't already been presented or outlined in the multitude of papers leveraging RL for Sepsis, and healthcare more broadly. Surveys are greatly needed in our community as far as they are thorough and detailed. However this paper barely gets beyond surface level interrogation of the literature and misses significant work produced broadly. I was disappointed that none of the work from Joseph Futoma (especially his work with Mark Sendak which actually put an RL system into clinical use for sepsis treatment!), Shengpu Tang, Taylor Killian, Suchi Saria, and many many others was not considered.

Ultimately, I did not feel that this paper was near to the quality to where I would begin to consider it for publication. There was no real novelty introduced beyond a very basic speculative framework to where agentic AI was discussed. The paper does not feature any actual conceptual development, analytical formulation, or any other foundation from which to judge this as a scholarly contribution.

**Questions:**

I do not have any questions for the authors. As there were no real technical contributions produced in this work, I do not have any further need to identify areas that I misunderstood or seek clarification to better appreciate what was presented.

---

### Meta-Review · Area_Chair_YYjt · 2025-12-08

**Summary:**

This paper was reviewed by 3 knowledgeable reviewers who raised concerns about:
1. Novelty (DkcG, 7oJs, R5AV): an agentic AI framework for sepsis care, with unclear positioning w.r.t. existing work.
2. Missing relevant literature (DkcG, 7oJs, R5AV).
3. Claims not well supported by empirical nor theoretical evidence (7oJs).
4. Unclear motivation for agentic AI (R5AV).

**Reviewer Concerns:**

Unfortunately, there was no rebuttal, so all the reviewers' concerns remain.

**Reviewer Scores:**

There was no rebuttal, so scores were not impacted.

---

### Decision · Program_Chairs · 2026-01-26

Reject